# Robust Control of Repeated Drug Administration with Variable Doses Based on Uncertain Mathematical Model

**DOI:** 10.3390/bioengineering10080921

**Published:** 2023-08-03

**Authors:** Zuzana Vitková, Martin Dodek, Eva Miklovičová, Jarmila Pavlovičová, Andrej Babinec, Anton Vitko

**Affiliations:** Institute of Robotics and Cybernetics, Faculty of Electrical Engineering and Information Technology, Slovak University of Technology in Bratislava, 841 04 Bratislava, Slovakia; zuzana.vitkova@stuba.sk (Z.V.); eva.miklovicova@stuba.sk (E.M.); jarmila.pavlovicova@stuba.sk (J.P.); andrej.babinec@stuba.sk (A.B.); anton.vitko@stuba.sk (A.V.)

**Keywords:** pharmacokinetics, compartmental models, closed loop control, repeated drug administration, robust control

## Abstract

The aim of this paper was to design a repeated drug administration strategy to reach and maintain the requested drug concentration in the body. Conservative designs require an exact knowledge of pharmacokinetic parameters, which is considered an unrealistic demand. The problem is usually resolved using the trial-and-error open-loop approach; yet, this can be considered insufficient due to the parametric uncertainties as the dosing strategy may induce an undesired behavior of the drug concentrations. Therefore, the presented approach is rather based on the paradigms of system and control theory. An algorithm was designed that computes the required doses to be administered based on the blood samples. Since repeated drug dosing is essentially a discrete time process, the entire design considers the discrete time domain. We have also presented the idea of applying this methodology for the stabilization of an unstable model, for instance, a model of tumor growth. The simulation experiments demonstrated that all variants of the proposed control algorithm can reach and maintain the desired drug concentration robustly, i.e., despite the presence of parametric uncertainties, in a way that is superior to that of the traditional open-loop approach. It was shown that the closed-loop control with the integral controller and stabilizing state feedback is robust against large parametric uncertainties.

## 1. Introduction

The processes running in biological systems, such as in a cell, an organ, or the whole living organism, are very complex, and their dynamics depend on the actual internal states of the system and their interactions; hence, they belong to the category of dynamic systems. In a cell, chains of biochemical reactions typically occur, which can be analyzed and modelled using discrete automata or Petri nets. The main approaches to describe the processes running in the body include the ordinary or partial differential equations, which are abstract mathematical models of the processes. Therefore, the mathematical modeling of the processes in bioscience plays a decisive role in understanding their characteristics and behaviors, which are often hidden. The modelling and control of these processes are based on theoretical results of biocybernetics.

The transport of the drug throughout the body is influenced by a chain of mutually connected and often not fully understood macro and micro processes. Therefore, we usually have significant variability in the choice of modelling structures. A typical uncertainty results from choosing an inadequate structure, due to which some important subprocesses are neglected. This represents the case of the so-called unmodelled dynamics [1].

Another source of uncertainty arises from inaccurate information about the values of model parameters. These include, for instance, the rate constants of the drug absorption from the site of administration to the blood circulation, and the rates of the drug exchange between the body compartments, among others. The mechanism described above is usually referred to as a parametric uncertainty.

The presence of uncertainties leads to their incorporation into the model and in the control design. For that reason, controlled drug delivery, which is required to ensure an optimal therapeutic effect, cannot be resolved solely via experimental sciences. Simply said, the uncertain of in vitro and/or in vivo models naturally calls for an application of cybernetic (system-based) methodologies.

The novelty of the solution presented in this paper is the design of the robust discrete time feedback control system, which can ensure that the drug concentration follows the desired steady state despite the uncertain parameters of the system.

Such applications of system-based approaches to solve problems related to drug kinetics can be found in [2,3]. Additionally, other relevant results have been presented in our recent work [4,5]. In [4], the influence of the surfactant monolaurin of sucrose (MLS) on the rate of absorption was analyzed after an instantaneous per os (peroral) administration to rats of sulfathiazole in the form of suspension. From the in vivo samples of the drug concentrations, the pharmacokinetic model was identified, which was later used to predict the absorption rate constant. The questions related to the role of auxiliary substances in pharmaceutical technology can be found in [6,7].

Based on the available sources and the experiments performed by the authors, the magnitudes of the model uncertainties were assessed and incorporated into the model. Therefore, following the results of [4], in [5], the state-bounding observer was designed for the uncertain model. The observer predicted the guaranteed upper and lower limits of possible drug concentrations in the body compartments, particularly those that are inaccessible for the collection of drug samples. Due to the observer, it was sufficient to take blood samples only from the tissue compartment, and the observer predicted whether the drug concentrations in the other compartments would violate the therapeutic range. The functionality of the observer was demonstrated considering three different models with physiology-based structures. The general theory of state observers can be found in [8,9,10,11,12].

This paper can be considered an extension of the previous results. It attempts to solve the problem of determining the sizes of individual doses that would be sufficiently flexible with respect to the specific requirements of therapy. The requirements also include adaptation of the dosing protocol to the required rate of change in the drug concentration in the body to a desired level and to subsequently maintain it at this level despite uncertain and possibly unstable system dynamics.

There exist various methods to evaluate the influences of the deterministic/random factors and to assess the suitability of the modeling structure for a particular purpose. To mention a few, there are population-based pharmacokinetic models (popPK), linear and nonlinear mixed-effect models (NLME), and physiology-based models (PBM). These also involve methods for determining the suitability of a model, such as Akaike’s or Schwarz’s information criteria (AIC or SC) [13].

## 2. Materials and Methods

The in vivo experiment is described in detail in [4,14]. In a nutshell, the experiment was carried out as follows: In the in vivo experiment, 36 rats with a weight of 200 g were considered, which were divided into six groups. Water suspensions containing 5% sulfathiazole were prepared, and at the beginning, a dose of 0.5 mL of suspension per 100 g was administered to all 36 rats. After six hours, using the heparinized injection syringe with cannula, a 2 mL sample of blood was withdrawn from the first group of six animals. The sampling was repeated every hour, but each time for another six animals because the previous group of rats was already dead. Hence, the second group was sampled seven hours after administration, the third group was sampled after eight hours, etc. Then, the mean values of the six samples obtained from each of the six groups were calculated.

In this way, the time dependence of the drug concentrations [mg/mL] versus time [h] was obtained as documented in Table 1. The choice of suspension as a dosage form resulted from the poor solubility of sulfathiazole in water.

While the value of administered dose is always exactly known, the state variables (concentrations in the compartments) are uncertain to a non-negligible extent due to uncertain parameters. Therefore, the problem calls for a system-based methodology, in particular robust control. To this end, a generalized compartmental model with the parameter Δ determining the uncertainty of the parameters is synthesized and the process is robustly stabilized using the specially designed state feedback.

## 3. Results

### 3.1. Continuous Time Model

The method used in this paper is based on the compartmental model shown in Figure 1. The model was parametrically identified using the least squares method.

Let us recall that the general form of the linear system with single input and single output (SISO) is described by the following system of n first-order differential equations:(1)x˙=Axt+but,yt=cTxt,
where xt∈ℝn×1, b∈ℝn×1, and c ∈ℝn×1 are the state, control, and observation vectors, respectively, and A∈ℝn×n is the system matrix, while yt∈ℝ and ut∈ ℝ are the output and input of the system, respectively. Generally, if A is a Metzler matrix and vector b ≥ 0, system (1) belongs to the category of linear positive systems. In addition to that, if A is Metzler and Hurwitz, then system (1) is positive and stable. Specifically, if a positive and stable system respects the rule of mass balance, it is called compartmental [4]. Note that linear and nonlinear compartmental systems are essential for describing the transport of an administered drug throughout the body. In particular, the three-compartment system shown in Figure 1 will be the subject of further analysis in this paper. Its mathematical model takes the following form [5]:(2)x˙1tx˙2tx˙3t=−ka+ke100ka−k2300k23−ke3x1tx2tx3t+100utyt=001xt

Parameters $k_{a}$, $k_{e1}$, $k_{23}$, and $k_{e3}$ $[h^{-1}$] are real positive constants. Parameter $k_{a}$ quantifies the rate of the drug absorption from the site of its administration. In this case, it is the gastrointestinal tract (abbr. GIT). The values of $k_{e1}$, $k_{e3}$ are the rates of the drug elimination from the compartments $x_{1}$ and $x_{3}$, and $k_{23}$ is a measure of the rate of the drug flow from $x_{2}$ to $x_{3}$.

The relation between the system input $u\left(t\right)$—the administered doses—and the system output $y\left(t\right)$—the observed concentrations of the drug in the body—is given by the following transfer function [4]:(3)$$G\left(s\right)=\frac{y\left(s\right)}{u\left(s\right)}=c^{T}{\left(sI-A\right)}^{-1}b,$$
where “s” indicates that $y\left(s\right)$ and $u\left(s\right)$ are the Laplace transforms of signals $y\left(t\right)$ and $u\left(t\right)$ [15,16]. The following nominal parameters were identified in [5]: $k_{a}=0.0370{\;h}^{-1}\;k_{e1}=0.1214{\;h}^{-1}\;k_{23}=1.2725{\;h}^{-1}\;k_{e3}=0.2171{\;h}^{-1}$.

**Remark 1.** *After performing the mathematical operations indicated on the right side of (3), we obtain the so-called system transfer function *$G\left(s\right)$, *which defines the relation between the system input and output. It can be expressed in the form of a faction of two polynomial functions (4)*.
(4)$$G\left(s\right)=\frac{y\left(s\right)}{u\left(s\right)}=\frac{b_{m}s^{m}+b_{m-1}s^{m-1}+\ldots +b_{0}}{a_{n}s^{n}+a_{n-1}s^{n-1}+\ldots +a_{0}}$$*The polynomial in the denominator is the so-called characteristic polynomial and can be used to determine the system stability. Parameters a_i_ and b_i_ include the rate constants* $k_{a}$, $k_{e1}$, $k_{23}$, and $k_{e3}$.

In contrast to the state (or internal) quantities $x_{1}$, $x_{2}$, $x_{3}$, which are hidden, the values of input *u*(*t*) and output *y*(*t*) can be directly measured. Therefore, while the parameters *a_i_, b_i_* are often easily identifiable, the situation for the constants $k_{a}$, $k_{e1}$, $k_{23}$, and $k_{e3}$ is more complicated. If we cannot determine them from the known values of *a_i_*, *b_i_* unambiguously, the system is considered “unidentifiable” and the values of the $\;k_{a}$, $k_{e1}$, $k_{23}$, $k_{e3}$ must be identified directly from the in vivo samples while considering the system (2). The details can be found in [17,18] and the references therein.

A detailed explanation of the biological meanings of the system parameters, together with their method of identification from the in vivo samples, can be found in [4,5].

Clearly, if the off-diagonal entries of the system matrix A in (2) are non-negative (Metzler matrix), the system (2) is called a positive system. In addition to that, for the nominal parameters, if every column-wise sum is non-positive, the model (1) satisfies the mass balance condition, implying that the system (2) is a nominally stable compartmental system. However, the situation may change when the parameters are subject to uncertainties. Therefore, we check the system stability based on the characteristic polynomial (5).
(5)$$\det\left(sI-A\right)=\det\left(\begin{array}{ccc} s+k_{a}+k_{e1} & 0 & 0\\ k_{a} & s+k_{23} & 0\\ 0 & -k_{23} & s+k_{e3} \end{array}\right)\;=\left(s+k_{a}+k_{e1}\right)\left(s+k_{23}\right)\left(s+k_{e3}\right)$$

The direct result of the above polynomial root decomposition shows that all three roots $\left(s+k_{a}+k_{e1}\right)$, $\left(s+k_{23}\right)$, $\left(s+k_{e3}\right)$ of this characteristic polynomial are negative for any $k_{a}>0$, $k_{e1}>0$, $k_{23}>0$, $k_{e3}>0$; hence, the nominal system is stable. This directly implies that the system with uncertain parameters, namely $k_{a}\pm \Delta k_{a}$, $k_{e1}\pm \Delta k_{e1}$, $k_{23}\pm \Delta k_{23}$, $k_{e3}\pm \Delta k_{e3},$ will remain stable if Δ>−1. The opposite case would be conditional to the existence of a negative parameter, which is biologically impossible.

### 3.2. Discrete Time Counterpart of the Continuous Time Model

The discrete time form of the model seems to be more suitable for the problem of finding an appropriate series of repeated drug doses (appropriate protocol). Specifically, for the continuous time system (1), the discrete time counterpart is given by the following difference equation:(6)xk+1=Fxk+guk,
where g∈ℝn×1 is the control vector and F∈ℝn×n is the state transition matrix.

Regarding Figure 1, the system output $y\left(k\right)$ is given by (7).
(7)$$y\left(k\right)=\underset{c^{T}}{\underbrace{\left(\begin{array}{ccc} 0 & 0 & 1 \end{array}\right)}}x$$

Unlike the continuous time model (1), the time t is now represented discretely as a sequence of the time instants t=kT, where T is the dosing period. In the in vivo experiment described in [4,14], the dosing period is equal to T=6 h and the same is considered in this paper. The relationship between matrix A of the continuous time system, matrix F of the discrete time system and between vectors b and g is given by (8) [17].
(8)$$\begin{array}{c} F=e^{AT}g=A^{-1}\left(F-I\right)b\\ g=A^{-1}\left(F-I\right)b \end{array}$$

In an analogy with the continuous time system (1), the discrete time system (6) is positive and stable if the matrix F is non-negative and Schur. Then, considering the dosing interval T=6 h from (8), we can obtain the nominal parameters given in (9).
(9)$$F=\left(\begin{array}{ccc} 0.3866 & 0 & 0\\ 0.0128 & 0.0005 & 0\\ 0.0718 & 0.3272 & 0.2718 \end{array}\right)\;g=\left(\begin{array}{c} 3.8726\\ 0.1025\\ 0.2704 \end{array}\right)$$

The steady-state gain [17], that is the ratio of the steady-state output $y_{ss}$ and the corresponding steady-state input $u_{ss}$ of system (6), equals the following:(10)$$c^{T}{\left(I-F\right)}^{-1}g=1.0759$$

### 3.3. Uncertain Discrete Time Model

Thus far, we have supposed that the entries of matrix F and vector g are known exactly, but their actual values can vary within some intervals around the nominal values. Consider the expected maximum acceptable parametric uncertainty Δ=± 0.1 that is ±10% of their nominal values. Then, the maximal system dynamics will be characterized by matrix $\bar{F}$ and vector $\bar{g}$, as follows:(11)$$\bar{F}=\left(\begin{array}{ccc} 0.4252 & 0 & 0\\ 0.0141 & 0.0005 & 0\\ 0.0789 & 0.3599 & 0.2990 \end{array}\right),\;\bar{g}=\left(\begin{array}{c} 4.2598\\ 0.1128\\ 0.2974 \end{array}\right),$$
and the minimal dynamics will be characterized by matrix $\underline{F}$ and vector $\underline{g}$, as follows:(12)$$\underline{F}=\left(\begin{array}{ccc} 0.3479 & 0 & 0\\ 0.0115 & 0.0004 & 0\\ 0.0646 & 0.2944 & 0.2446 \end{array}\right),\;\underline{g}=\left(\begin{array}{c} 3.4853\\ 0.0923\\ 0.2434 \end{array}\right).\;\underline{F}\leq F\leq \bar{F}\;\mathrm{and}\;\underline{g}\leq g\leq \bar{g}$$

Using the Kharitonov theorem [18], it can be easily shown that the uncertain system with the limiting matrices (11) and (12) is stable.

Now, having described the nominal, maximal, and minimal dynamics, we can find the answer to the principal question: Which trajectory $u\left(k\right),\;k=0,1,2,\ldots$ of the repeated drug doses is able to force the output trajectory $y\left(k\right)$ to converge to the requested steady-state value $y_{ss}=r$? In general, r is an arbitrary positive real number. Due to the presence of uncertainties, the answer to this question is not straightforward. There are two principal approaches, namely the drug dosing in the open loop and in the closed loop. Both approaches will be briefly explained in the following sections.

### 3.4. Drug Dosing in the Open Loop

In this strategy, the steady-state value of y, i.e., $y_{ss}$ for the unit input $u\left(k\right)=1$*,*
k=0,1,2 …∞ will be determined first. Then, based on $y_{ss}$, an appropriate dose $u\left(k\right)$ will be adjusted such that $y_{ss}$ should be equal to the requested value *r.* Formally, it can be described as follows:(13)$$\underset{k\to \infty }{\lim}\;y\left(k\right)=y_{ss}=r$$

Hence, the main idea behind the design of the open-loop dosing protocol is as follows: Consider the discrete time model of a general subject shown in Figure 1 and described by the set of difference Equations (6) and (7). Its input $u\left(k\right)$ is a sequence of the administrated doses and the output $y\left(k\right)$ is the corresponding sequence of the drug concentrations in the body, as shown in Figure 2.

Following the system theory [17], the steady-state value $y_{ss}$ is given by relation (14).
(14)$$r=y_{ss}=\underset{\mathrm{steady}-\mathrm{state}\;\mathrm{gain}}{\underbrace{\left[c^{T}{\left(I-F\right)}^{-1}g\right]}}u\left(k\right)$$

Note that the expression for the steady-state gain (denoted by the brace in (14)) is nothing more than a scalar multiplier by which the constant unit sequence $u\left(k\right)=1$*,*
k=0,1,2,3,… of the drug doses should be multiplied to obtain r. Therefore, for a given r and the steady-state gain defined using (14), the following constant drug doses should be repeatedly administered:(15)$$u\left(k\right)=\frac{r}{c^{T}{\left(I-F\right)}^{-1}g}$$

Note that we will consider the requested steady-state concentration r=50 mg/mL and the sampling period T=6 h. Therefore, the repeated constant doses $u\left(k\right)$ should be equal to the following:(16)$$u\left(k\right)=\frac{r}{c^{T}{\left(I-F\right)}^{-1}g}=\frac{50}{1.0759}=46.4727\;\mathrm{mg}.$$

For each of the experiments presented further, the area under curve (abbr. AUC) will be determined to quantify the total drug exposure across time considering the experiment length 144 h, nominal models, and using the trapezoidal rule.

The main drawback of this open-loop approach is that the doses are constant in size while presenting no feedback on the current drug concentration in the body. In other words, for a given r, the values of $u\left(k\right)$ remain constant regardless of the current drug concentration $y\left(k\right)$. Therefore, at least four disadvantages of this approach can be highlighted as follows:

First, the doctor has no means through which they could speed up or slow down the transition process, i.e., the process of gradually approaching the steady-state value $y_{ss}=r=50$ mg/mL. Second, the transition process of $y\left(k\right)$ is unique as it cannot be modified by the constant input sequence $u\left(k\right)$. Therefore, there are cases where the transition process can last for an unacceptably long time. The third and most serious drawback is that the behavior of the system induced by the uncertainties cannot be automatically compensated for.

The trajectories of drug concentration $y\left(k\right)$ for the calculated constant doses $u\left(k\right)=46.4727$ mg administered every 6 h during the period of 144 h are shown in Figure 3. It can be clearly seen that the nominal trajectory of $y\left(k\right)$ (full curve) reaches the requested value r=50 mg exactly and it is maintained within the therapeutic range. Contrary to that, assuming the effect of ±10% uncertainties of the nominal F and g, the steady-state value $y_{ss}$ leaves the therapeutic range (dotted and dashed curves). This means that for model (6), the parametric uncertainties ±10% are too large; hence, the open-loop therapy can significantly jeopardize the patient’s health. This is the fourth serious drawback of the open-loop dosing strategy.

Since the doctor is not aware of the extent of the parametric uncertainties, the aforementioned drawbacks naturally require a more sophisticated dosing. Much better management of the dosing protocol can be achieved using the closed-loop approach described in the Section 3.5.

### 3.5. Drug Dosing in the Closed Loop

In this approach, doses $u\left(k\right)$ are not constant as they will depend on the current samples of the drug concentrations $y\left(k\right)$, for k=0,1,2…. As shown by the closed-loop arrangement, the concentrations $y\left(k\right)$ will always converge towards the requested value r regardless of the uncertainties. Due to this property, one can claim that this control strategy is robust with respect to parametric uncertainties [19,20,21]. The main idea is illustrated in the scheme shown in Figure 4.

The block “subject” represents an ill body, which should be treated by repeatedly administrating non-constant doses $u\left(k\right)$. To this end, the control algorithm (17) is based on the so-called integral controller. The output of this controller is the sequence of doses $u\left(k\right)$, which are to be administered consecutively at the time instants t=kT for k=0,1,2,… [16]. As mentioned above, the role of $u\left(k\right)$ is to force the drug concentration $y\left(k\right)$ to converge towards the desired steady-state value $y_{ss}=r=50$ and to maintain it on this level despite the presence of parametric uncertainties.

From the physical and biological nature of the problem, it follows that the quantities $y\left(-1\right)$, $u\left(-1\right)$, $y\left(0\right)$ are equal to zero. The integral controller works according to algorithm (17).
(17)$$u\left(k\right)=u\left(k-1\right)+k_{i}\left[(r-y\left(k-1\right)\right]\;\;\mathrm{for}\;k=0,\;1,2,3,r=50\;\mathrm{mg}$$

The whole process works as follows: At the beginning, the tunable parameter $k_{i}$ (the so-called integral gain) [16] is set to some value, for example, $k_{i}=0.3$. This value defines not only the rate of increasing the subsequent doses $u\left(k\right)$, but also the number of doses required to reach the steady-state concentration $y_{ss}=r$. Then, for k=0, the initial dose is computed as $u\left(0\right)=0+0.3\left(50-0\right)=15$ mg and the doctor will administrate it. At the end of the first dosing interval (T=6 h), the doctor takes the first sample $y\left(1\right)$ and, together with the required steady-state value $y_{ss}=r$, passes this information to the controller (algorithm (17)), which determines the next dose as $u\left(1\right)=u\left(0\right)+0.3\left[50–y\left(0\right)\right]=15+0.3\left(50-4.05\right)=15+13.78=28.78$. The process is repeated for the next indices k. The simulated trajectory of $u\left(k\right)$*,* for k=0,1,…25 is shown in Figure 5.

The corresponding trajectory of the drug $\mathrm{concentrations}\;y\left(k\right)$ in the site where samples are taken (i.e., from the third compartment) is shown in Figure 6.

For illustration purposes, the first ten values of the computed doses $u\left(k\right)$ and the corresponding trajectory of the drug concentrations $y\left(k\right)$ for the nominal case (Δ=0) are given in Table 2.

As Figure 5 and Figure 6 illustrate, both trajectories $u\left(k\right)$ and $y\left(k\right)$ exhibit some overshoots above their steady-state values. This is caused by choosing an integration gain $k_{i}$ that is too high. Threfore, it is recommended to decrease its value to $k_{i}=0.1$. After carrying this out, more appropriate trajectories are obtained, as shown in Figure 7 and Figure 8.

The nominal trajectory is presented by the full line. The trajectory for positive uncertainties (11) is presented by the dashed curve, and the dot trajectory corresponds to the negative uncertainties (12).

It is even possible to speed up the transition process by intentionally applying a higher loading dose, such as $u\left(0\right)=50$ mg; by carrying this out, we achieved the desired results, as demonstrated in Figure 9. 

We can artificially disturb the control configuration and the experiment described above, first, by applying underestimated and, second, by applying overestimated initial doses to make the control algorithm properly correct the dosing. The corresponding response to drug concentration with the first five doses scaled by a factor of 0.1 can be seen in Figure 10, where we can essentially observe a delayed start of the treatment. On the contrary, the response of drug concentration with the first five doses scaled by a factor of 5 can be seen in Figure 11, demonstrating that overestimation can lead to deteriorated performance or even toxic treatment; nevertheless, the control algorithm could ultimately manage this scenario.

It is worth mentioning that from the aspect of system theory, the processes of the drug distribution through the body belong to the category of dynamic processes [15]. Therefore, they can be stable or unstable depending on the ways in which the subsystems are mutually connected, as well as the model parameters. The only source of instability of the control structure shown in Figure 4 can arise from an inappropriate (usually too high) integration gain $k_{i}$. This was documented by the overshoots in Figure 5 and Figure 6, indicating that the value $k_{i}=0.3$ is too high and its further increase may lead to the instability of the closed loop. For that reason, $k_{i}$ should be chosen to prevent the system from destabilization.

As an illustration, we will check the stability for the considered integral gain $k_{i}=0.1$. The transfer function form of the nominal discrete time model (6) and (7) obtains the following:(18)$$\frac{y\left(z\right)}{u\left(z\right)}=c^{T}{\left(I-Fz^{-1}\right)}^{-1}gz^{-1}$$

Then, we can derive the closed-loop transfer function as follows:(19)$$\frac{y\left(z\right)}{r\left(z\right)}=\frac{k_{i}\;c^{T}{\left(I-Fz^{-1}\right)}^{-1}g\;z^{-1}}{\left(1-z^{-1}+k_{i}c^{T}{\left(I-Fz^{-1}\right)}^{-1}gz^{-2}\right)}$$

For the nominal model with F and g given by (9), we have the following transfer function:(20)$$\frac{y\left(z\right)}{r\left(z\right)}=\frac{0.027z^{-1}+0.027z^{-2}+0.0003z^{-3}}{1-1.6319z^{-1}+0.7850z^{-2}-0.1051z^{-3}+0.0001z^{-4}}$$

The roots of its characteristic polynomial are $\left\{0.8643,\;0.5445,\;0.2226,\;0.0005\right\}$.

For Δ=0.1 with $\bar{F}$ and $\bar{g}$ given by (11), we have the following:(21)$$\frac{y\left(z\right)}{r\left(z\right)}=\frac{0.029z^{-1}+0.025z^{-2}+0.0004z^{-3}}{1-1.6950z^{-1}+0.8773z^{-2}-0.1272z^{-3}+0.0001z^{-4}}$$

The roots of its characteristic polynomial are $\left\{0.7699,\;0.6840,\;0.2406,\;0.0005\right\}$.

For Δ=−0.1 with $\underline{F}$ and $\underline{g}$ given by (12), we have the following:(22)$$\frac{y\left(z\right)}{r\left(z\right)}=\frac{0.024z^{-1}+0.0167z^{-2}+0.0002z^{-3}}{1-1.5687z^{-1}+0.6951z^{-2}-0.0852z^{-3}+0.00005z^{-4}}$$

The roots of its characteristic polynomial are $\left\{0.9037,\;0.4606,\;0.2039,\;0.0004\right\}$. Since the roots of all characteristic polynomials (20)–(22) lie in the unit circle, the closed-loop control is stable.

The optimal value of $k_{i}$ should be a trade-off between the speed of increasing the control response of $y\left(k\right)$ and the system stability. It is also important to note that, contrary to the open-loop case, the individual doses $u\left(k\right)$ are not constant; rather, they gradually increase while their increments decrease as $u\left(k\right)$ and $y\left(k\right)$ approach their steady-state values. This is evident from Figure 5 and Figure 7. In addition to that, these figures illustrate that at the time instant t=24T, the dose is $u\left(24\right)=46.27$ mg, which is virtually equivalent to the value determined for the open-loop control, i.e., 46.47 mg, from (16).

Another important observation is that Figure 5, Figure 6, Figure 7 and Figure 8 convincingly demonstrate the robustness of the closed-loop control, meaning that if the parametric uncertainties do not exceed their limits Δ, the system remains stable. Hence, contrary to the open loop, the closed-loop approach with the integral controller ensures that both in the nominal process and in the process disturbed by the parametric uncertainties, the drug concentration $y\left(k\right)$ converges to the requested steady-state value r=50 mg/mL, which implies that the control system is robust.

In addition to the advantages of closed-loop dosing mentioned earlier, another practical advantage is the possibility of changing the rate of increasing the trajectory $u\left(k\right)$ by changing the controller gain $k_{i}$. Clearly, the change in the rate of $u\left(k\right)$ is related to the changes in the size of its increments, which, in turn, influence the number of doses needed to reach the requested level r. To see this, compare Figure 5 and Figure 7.

### 3.6. Drug Dosing in the Case of an Unstable Subject

It was shown that the discrete time system (6) is stable for all positive parameters. On the other hand, some special models of bioprocesses, such as physiology-based pharmacokinetic, economic, and ecological models, may not share this feature. Even if their states of equilibria are stable, the region of stability around them can be very small. Therefore, even negligibly small parametric changes or exogenous factors affecting the living subject may render the system unstable. This is manifested by the limitless increase in some characteristic quantities, in particular the system states.

For example, imagine a model of tumor growth, which is typically defined by a set of differential equations [16]. The normally stable tumor-free state equilibrium, i.e., the one computed from the nominal model, can become unstable due to the deviation of the parameters from their nominal values. The instability of the equilibrium will manifest itself through some (or all) variables, e.g., the tumor volume will grow beyond all limits.

However, in this section, we will not deal with any specific model of tumor growth. Instead, we will artificially destabilize the pharmacokinetic model described by (6) and (7). It will be shown that the closed-loop control structure can also be used to control an unstable subject. Consider the feedback scheme shown in Figure 12.

The principle of stabilization via state feedback generally assumes all three components of the state vector $x\left(k\right)=\left[x_{1}\left(k\right),\;x_{2}\left(k\right),\;x_{3}\left(k\right)\right]$. Therefore, the vector $x\left(k\right)$ should contain information on the drug concentrations in all three compartments. As shown in Figure 12, the state vector $x\left(k\right)$ is repeatedly sensed and passed to the state feedback where it is converted to the sequence of stabilizing doses $u\left(k\right)$.

Obviously, taking drug samples from all three compartments is biologically infeasible; however, their values can be obtained without the need for direct measurement. The states can be generated by the state observer (we have synthesized it in [5], which requires taking only the samples of the output variable $y\left(k\right)).$ To illustrate this, consider the stable discrete time system described in Equations (6) and (7). To make it artificially unstable, we will intentionally modify the lower uncertainty matrix $\bar{F}$ by adding a destabilizing matrix, as in (23).
(23)$$\begin{array}{c} \underline{F}=\left(\begin{array}{ccc} 0.3479 & 0 & 0\\ 0.0115 & 0.0004 & 0\\ 0.0646 & 0.2944 & 0.2446 \end{array}\right)+\left(\begin{array}{ccc} 0.7 & 0 & 0\\ 0 & 0 & 0\\ 0 & 0 & 0 \end{array}\right)\\ =\left(\begin{array}{ccc} 1.0479 & 0 & 0\\ 0.0115 & 0.0004 & 0\\ 0.0646 & 0.2944 & 0.2446 \end{array}\right) \end{array}$$

The eigenvalues of matrix (23) are as follows:(24)$$\mathrm{eig}\left(\underline{F}\right)=\left\{0.2446,\;0.0004,\;1.0479\right\}$$

Because the eigenvalue 1.0479 is larger than one, it indicates that the dynamic system (6) with matrix (23) in (9) is unstable [15,17]. The trajectories $u\left(k\right)$ obtained using (17) are shown in Figure 13.

It can be observed that for the parametric uncertainty Δ=−0.1, the dotted trajectory uk decreases and, at a certain time, becomes negative. Clearly, it can be concluded that the system does not work properly. Therefore, the system should be stabilized via state feedback as proposed. Stabilization was performed by coupling the subject with the stabilizing state feedback uk=Kxk. To determine *K*, we suppose the interval uncertainties included in (2) and apply Theorem 5 from [22]. To satisfy the conditions of this theorem, we resolved the corresponding problem (25) of linear programming [23].
(25)K=K++K−K+=100σ11+σ12+σ13+σ21+σ22+σ23+σ31+σ32+σ33+α1rTG_TvK−=100σ11−σ12−σ13−σ21−σ22−σ23−σ31−σ32−σ33−1rTG_Tvwhereσij+<ϵ+ ∀ i=1,2,3, j=1,2,3 σij−<ϵ− ∀ i=1,2,3, j=1,2,3 F¯T−Iv1v2v3+ϵ+ϵ+ϵ++ϵ−ϵ−ϵ−<000αF_ 1rTG_Tv+G_σ11+σ12+σ13+σ21+σ22+σ23+σ31+σ32+σ33++ αG¯σ11−σ12−σ13−σ21−σ22−σ23−σ31−σ32−σ33−≥000000000chosen  ϵ−<0chosen ϵ+>0chosen v>0,  v∈ℝ3×1chosen α>1 1r=111G_≡g_0

The obtained feedback gain vector K is as follows:(26)$$K=\left(\begin{array}{ccc} -0.2371 & 0.0754 & 0.0754 \end{array}\right)$$

The state feedback with the gain K stabilizes the closed-loop system in Figure 12 robustly, that is, under the conditions of uncertain entries of F and g. Actually, by applying the state feedback $u\left(k\right)=Kx\left(k\right),$ all eigenvalues of the closed-loop system shown in Figure 12 are smaller than one, including the following:(27)$$\mathrm{eig}\left(\underline{F}+gK\right)=\left\{-0.0054,\;0.1774,\;0.2309\right\}$$

Although all components of vector $x\left(k\right)$ are positive, it can be deduced from (26) that one component of the vector K is negative. This means that for some special values of the components of $x\left(k\right),$ the product $Kx\left(k\right)$ can result in negative doses $u\left(k\right),$ which is infeasible. Therefore, the scheme in Figure 12 cannot be used alone. It must be part of the closed loop in Figure 4, as shown in Figure 14. Then, the requested steady-state value r can be flexibly set up to an arbitrary positive value.

Although the original subject (without state feedback) is unstable, the inner closed-loop system shown in Figure 14 is stable. The resulting drug doses $u\left(k\right)$ are given by the sum of the controller output $v\left(k\right)$ and the feedback signal $fs\left(k\right)$ is as follows from (28).
(28)$$u\left(k\right)=v\left(k\right)+fs\left(k\right)=v\left(k\right)+K\;x\left(k\right)$$

The first ten values of the feedback signal $fs\left(k\right)$ and the drug doses $u\left(k\right)$ for T=6 h and $k_{i}=0.3$ are given in Table 3 and Table 4. While the samples of the feedback signals $fs\left(k\right)$ are negative, the doses $u\left(k\right)$ are positive for any positive value of r. The corresponding trajectories of $y\left(k\right)\equiv x_{3}$ are shown in Figure 15.

Although the trajectories $y\left(k\right)$ displayed in Figure 6 and Figure 15 correspond to the same dosing period and the controller gain (T=6 h, $k_{i}=0.3$), they are not quite equivalent. This was expected because, in the previous case, algorithm (16) controlled the stable subject alone, whereas now it controls the subject augmented by the stabilizing state feedback; therefore, it has different dynamics. Regardless of this, by suitably adjusting the integral gain $k_{i},$ the shape of the trajectory $u\left(k\right)$ can be affected as before.

## 4. Discussion

The open-loop approach with the dose determined offline can be seen to be theoretically correct and was demonstrated to work properly under the idealized assumption of model parameters that are exactly known. However, it was not possible to prevent the situation where the steady-state concentration of the drug is outside the therapeutic range due to parametric uncertainties. The parametric uncertainties ultimately caused a bias in the system static gain, rendering the determined open-loop dose either ineffective or toxic. Therefore, such a design cannot be considered robust [19,20].

On the other hand, the advantage of the closed-loop approach with the integral controller is that the drug concentrations converged to the requested steady-state value regardless of the actual values of uncertain model parameters. Due to this, for an appropriately chosen controller gain ($k_{i}$), the trajectories of the drug concentrations converged to the desired steady-state level (r) aperiodically (i.e., without overshoots). Therefore, any violation of the therapeutic range could be avoided and the therapy was completely safe.

Finally, by adding the state feedback control with an appropriate robust control design, we could achieve the stabilization of a system that became unstable due to uncertainties while ensuring the convergence of the drug concentration to the requested steady-state value. It was shown that after stabilization of the unstable object, it was possible to successfully control the repeated drug administration.

Compared with the fixed dose protocol, the closed-loop delivery scheme is very flexible in affecting the time until the drug concentration reaches the therapeutic range. The performance of the closed loop therapy can be tuned by adjusting the integration gain $k_{i}$ and the state-feedback gain K in terms of tuning the corresponding parameters of robust control design algorithm (25), which should be chosen in such a way that the monotonic concentration growth is ensured. In this way, the aggressiveness of the therapy can be adjusted to minimize the delays in reaching the therapeutic range by increasing the integration gain. However, a reasonable trade-off between the speed and safety of the therapy must be chosen since more aggressive policies usually lead to dangerous overshoots and, ultimately, to deteriorated robustness. It is important to note that with increasing control aggressiveness, the sensitivity of the control performance with respect to the magnitude of parametric uncertainties also typically increases.

## 5. Conclusions

This paper considers the pharmacokinetic compartmental model, which the authors designed and parametrically identified from in vivo concentration samples [4,14]. The aim was to design a dosing protocol for repeated administration. It was supposed that the model parameters are uncertain due to numerous reasons, which means that their values vary within a finite interval. The limits of the parametric uncertainties were individually set to 10% of the corresponding nominal values.

Because repeated drug dosing is essentially a discrete time process, it was decided to solve the problem completely in the discrete time domain. To this end, the original continuous time model was first transformed into its discrete time counterpart, and both the open-loop and the closed-loop analyses were performed solely in the discrete time domain. The repeated drug dosing problem was approached in the following three different ways:

First, in the open-loop approach, the system static gain and the doses of constant size, which were repeatedly administered, were determined, forcing the drug concentration to reach the given steady-state level of the drug concentration and to maintain this level for a given time interval.

Second, in the closed-loop approach, the initial oral dose was administered, and after the dosing interval T=6 h, the sample of the drug concentration was taken. Information about its value was passed to the integrating controller, which computed the size of the next dose. This strategy could ensure that drug concentrations converge to the requested steady-state value regardless of actual uncertain model parameters.

In the third approach, the problem of repeated drug dosing for the subject (more precisely, its model), which was not only uncertain but also unstable, was resolved. Although, at first glance, such a situation may seem very rare or even impossible to happen in a living body, it has been shown that this is nothing unnatural. These issues were discussed in Section 3.6.

It can be concluded that the main practical limitation of the proposed strategy is in providing full state feedback; yet, this requirement can be reduced using a state observer [5]. Hence, the required feedback reduces to sensing only the drug concentration in the output compartment, which, today, is feasible due to the extensive development of sensors and wearable electronic devices for biomedical applications.

The results presented in Section 3.6 create the basis for a future analysis of the possible treatment of tumor growth, which is typically an unstable system. This problem is briefly outlined in Appendix A in the form of a short authors’ reflection.

## Figures and Tables

**Figure 1 bioengineering-10-00921-f001:**
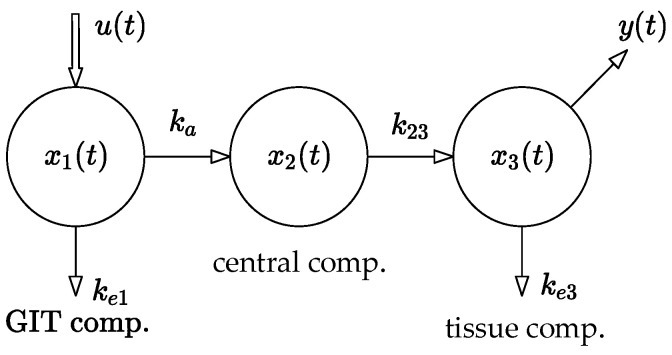
The considered three-compartment pharmacokinetic model showing the drug transport mechanisms, state variables, and model parameters. Variables $x_{1}$, $x_{2}$, $x_{3}$ are the drug concentrations in the gastrointestinal tract (abbr. GIT), central compartment, and tissue compartment, respectively, while u and y are the input and output of the model, respectively.

**Figure 2 bioengineering-10-00921-f002:**
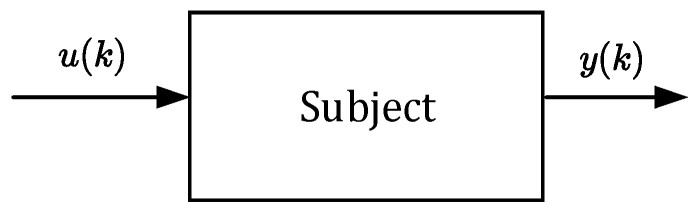
Cybernetic abstraction of a living organism (treated subject) in the case of open-loop dosing approach.

**Figure 3 bioengineering-10-00921-f003:**
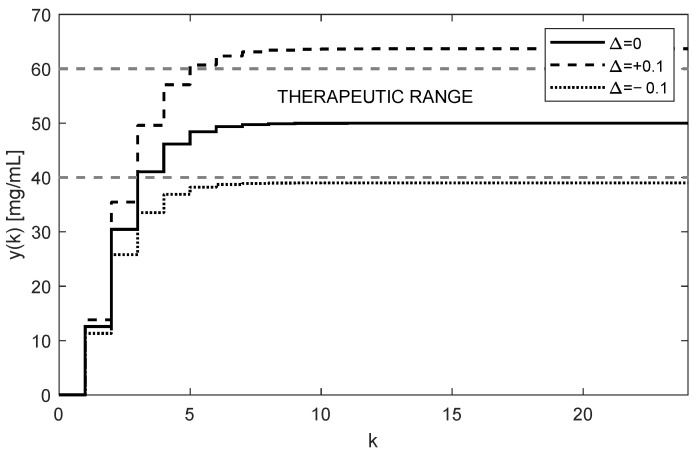
Trajectories of the drug concentrations $y\left(k\right)$ in the case of repeated constant dose showing poor robustness of the open-loop approach resulting in either ineffective or toxic treatment (AUC = 6615.7 h × mg/mL).

**Figure 4 bioengineering-10-00921-f004:**
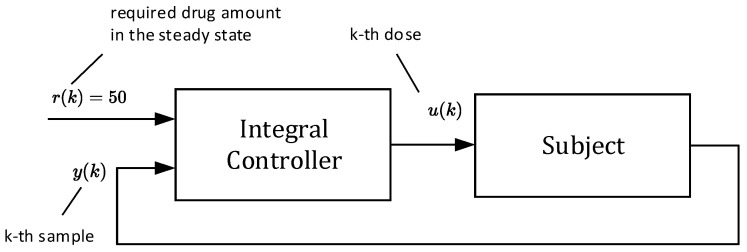
Block diagram of drug dosing in the closed loop that involves the integral controller with feedback on the drug concentration to determine the new dose size.

**Figure 5 bioengineering-10-00921-f005:**
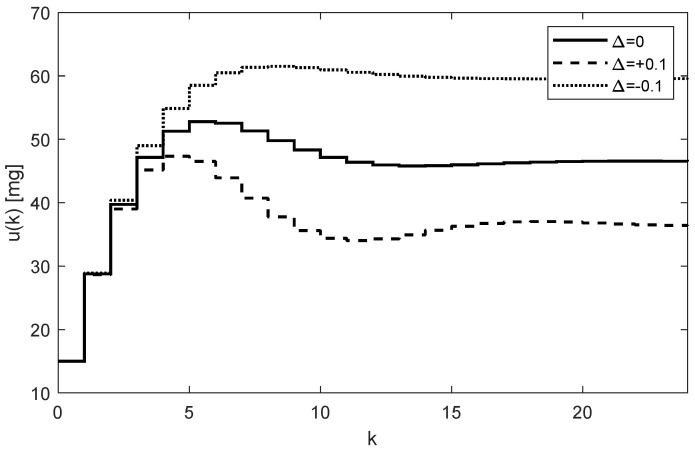
Trajectories of the drug doses $u\left(k\right)$ for the integral controller with $k_{i}=0.3$ showing a relatively fast response.

**Figure 6 bioengineering-10-00921-f006:**
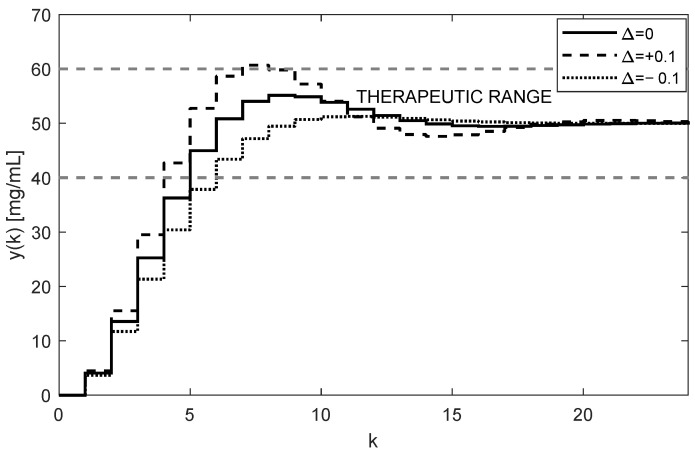
Trajectories of the drug concentration $y\left(k\right)$ for the integral controller with $k_{i}=0.3$ showing a relatively fast but robust response with slightly periodic behavior and overshoots (AUC = 6419.7 h × mg/mL).

**Figure 7 bioengineering-10-00921-f007:**
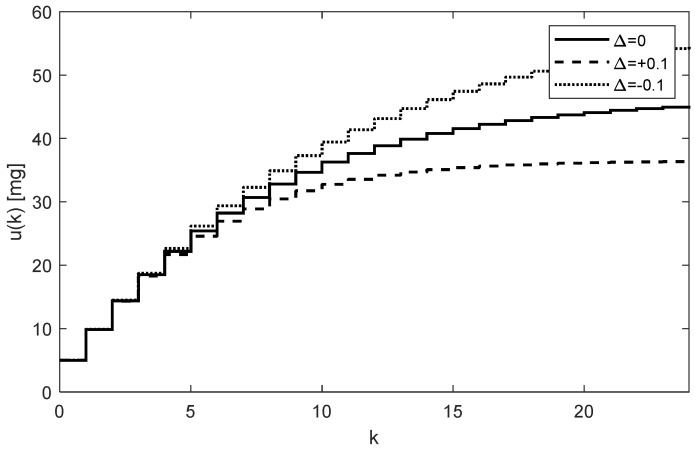
Trajectories of the drug doses $u\left(k\right)$ for the integral controller with $k_{i}=0.1$ showing a relatively fast response.

**Figure 8 bioengineering-10-00921-f008:**
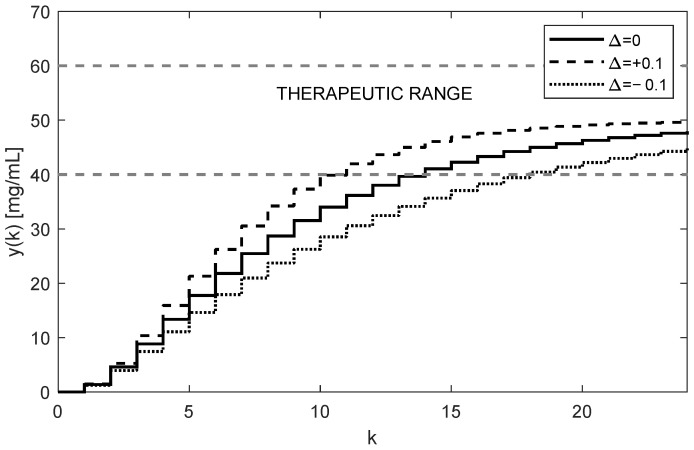
Trajectories of the drug concentrations $y\left(k\right)$ for the integral controller with $k_{i}=0.1$ showing a relatively slow but robust response with aperiodic behavior and no overshoots (AUC = 4647.5 h × mg/mL).

**Figure 9 bioengineering-10-00921-f009:**
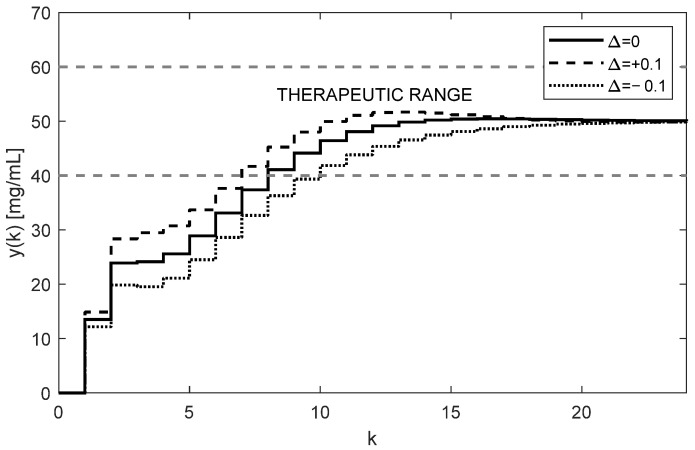
Trajectories of the drug concentrations $y\left(k\right)$ for the integral controller with $k_{i}=0.1$ and application of a high loading dose $u\left(0\right)=50$ mg (AUC = 5956.0 h × mg/mL).

**Figure 10 bioengineering-10-00921-f010:**
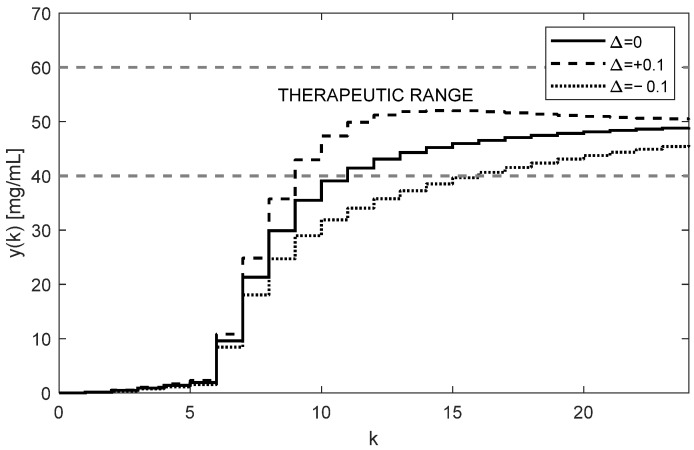
Trajectories of the drug concentrations $y\left(k\right)$ for the integral controller with $k_{i}=0.1$ and underestimated initial doses (AUC = 4605.0 h × mg/mL).

**Figure 11 bioengineering-10-00921-f011:**
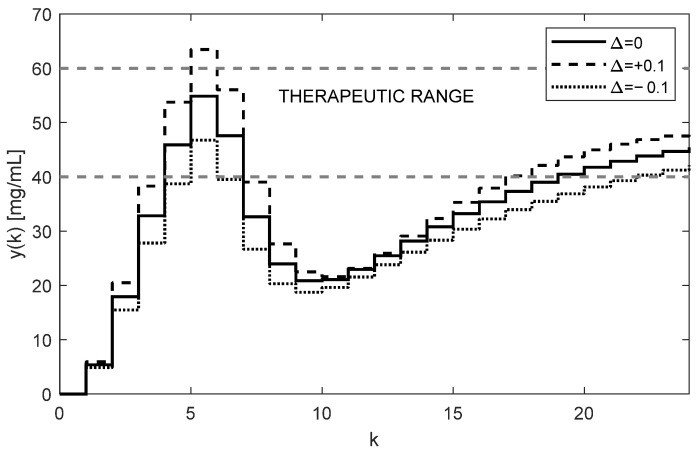
Trajectories of the drug concentrations $y\left(k\right)$ for the integral controller with $k_{i}=0.1$ and overestimated initial doses (AUC = 4751.2 h × mg/mL).

**Figure 12 bioengineering-10-00921-f012:**
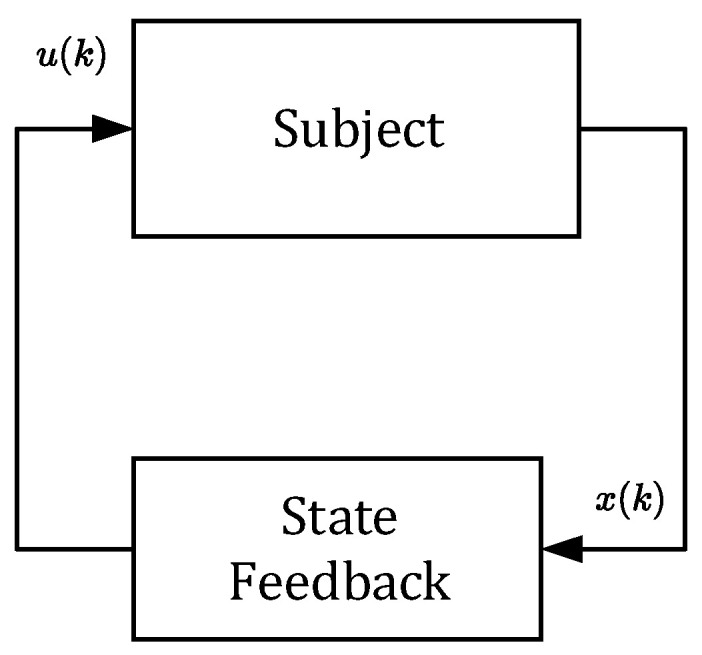
Block diagram of the inner stabilizing closed loop for the drug dosing that involves the state-feedback controller to determine the new stabilizing dose size.

**Figure 13 bioengineering-10-00921-f013:**
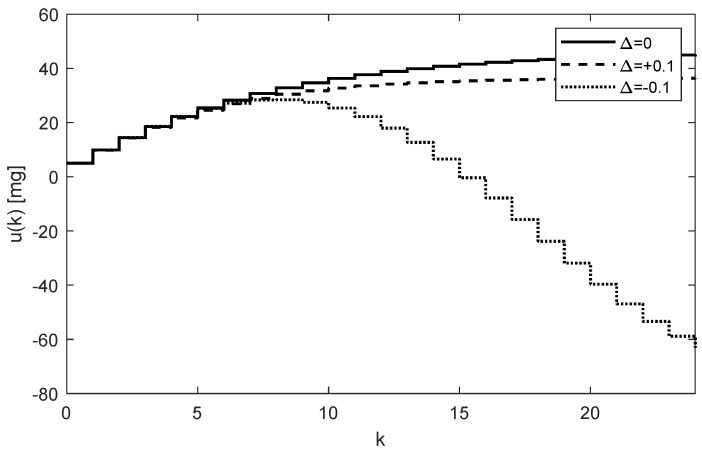
Trajectories of the drug doses $u\left(k\right)$ for an unstable subject showing unfeasible negative drug doses to be administered.

**Figure 14 bioengineering-10-00921-f014:**
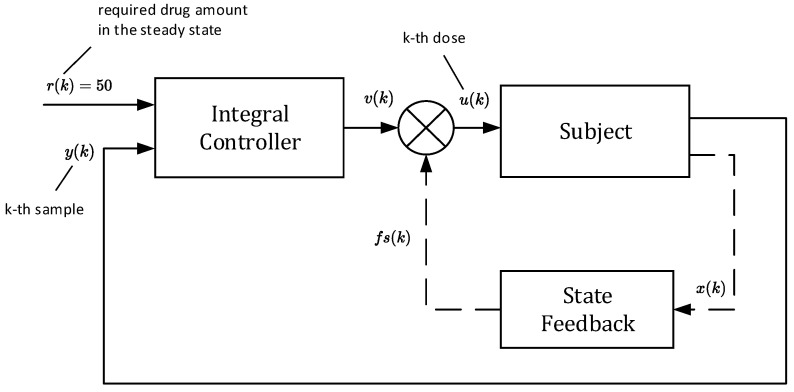
Block diagram of the cascade control loop for the robustly stabilizing drug dosing that involves the state-feedback controller and the integral controller to determine the new dose size.

**Figure 15 bioengineering-10-00921-f015:**
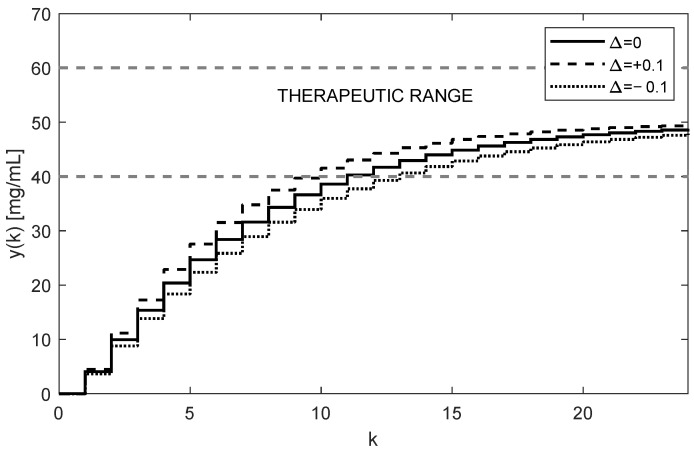
Robustly controlled drug concentrations $y\left(k\right)$ of an unstable subject stabilized by the state feedback showing stable behavior despite the unstable nature of the system (AUC = 3737.0 h × mg/mL).

**Table 1 bioengineering-10-00921-t001:** In vivo concentrations of the drug.

t [h]	0	1	2	3	4	5	6
c [mg/mL]	0	0.0715	0.0855	0.0780	0.0735	0.0490	0.0535

**Table 2 bioengineering-10-00921-t002:** Drug doses $u\left(k\right)$ and trajectory of drug concentrations $y\left(k\right)$ for the nominal case.

*k*	0	1	2	3	4	5	6	7	8	9
*u*(*k*)	15.00	28.78	39.71	47.14	51.26	52.77	52.52	51.31	49.77	48.30
*y*(*k*)	0	4.05	13.55	25.24	36.25	44.96	50.82	54.03	55.15	54.88

**Table 3 bioengineering-10-00921-t003:** Trajectory of the feedback signal $fs\left(k\right)$ for the nominal case.

*k*	0	1	2	3	4	5	6	7	8	9
***fs*(*k*)**	0	−3.02	−5.25	−7.53	−9.64	−11.65	−13.54	−15.32	−17.00	−18.58

**Table 4 bioengineering-10-00921-t004:** Trajectory of the drug dose $u\left(k\right)$ for the nominal case.

*k*	0	1	2	3	4	5	6	7	8	9
***u*(*k*)**	15.00	19.70	25.17	29.21	32.55	35.23	37.40	39.15	40.56	41.70

## Data Availability

The in vivo data are available from the first author.

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
