# Peer review of "Robust Control of Repeated Drug Administration with Variable Doses Based on Uncertain Mathematical Model"

_bioengineering, 2023, doi:10.3390/bioengineering10080921_

Round 1

Reviewer 1 Report

This is work the extension (continuation) of authors previous work. This study is worth publishing. But the authors must improve it. The methodology is not appropriately described, only referring to previous work is not enough. Table 1 and 2 can be combined. The captions of the figure are not self-explanatory. The conclusion of the entire manuscript is missing. Overall, the manuscript is poorly written and the whole manuscript should be rephrased and ambiguous statements should not be there e.g. Line 42 of page 1…….can be found in the scientific papers [2,3], and some can be found in the papers of the authors. 

Poorly written

Author Response

See the attached document

Reviewer 2 Report

1)  In such case we talk about parametric uncertainty. Change the sentence. 

2) Functionality of the observer was demonstrated by 57 three different, but physiologically substantiated model structures. This sentence is not clear. 

3) The reader can find out that the roots of this characteristic polynomial are negative for any ?? > 0, ??1 > 0, ?23 > 0, ??2 > 0. Check it.

4) Give one more reference here. Due to this property, one can claim that such control is robust with respect to the parametric uncertainties. 

5) Write separately the conclusion section. 

Minor changes are required. Check Grammar and English sentences. 

Author Response

See the attached document

Reviewer 3 Report

This MS presents an analytical model that shows how the use of a closed-loop control system might be used to control drug dosing were it possible to measure the concentration of a drug in at least one compartment of a multi-compartment system. It is written in a clear, pedagogically sound manner that provides sufficient detail that a serious student of pharmacokinetics could readily follow and implement the procedure as a discrete time process. Automated drug delivery, for example by a programmable infusion pump might make the approach practical, if it is shown possible to estimate the correct pharmacokinetic parameters. The authors should discuss how the controlled delivery scheme delays the time to the therapeutic range as compared to the open-loop, fixed dose protocol. The MS would be greatly improved with several more examples that would demonstrate the power of the closed-loop approach. 1) When the initial dose(s) are overestimated and the closed-loop algorithm has to correct the dosing. What is involved in making that correction promptly convergent. 2) When the initial dose(s) are significantly underestimated and the closed-loop algorithm has to correct the dosing. 3) Where the initial “loading” dose is intentionally higher and decreased rapidly with time so as to approach the therapeutic range in the shortest possible time without overshoot. Each of the analyses presented should include a computation of the temporal development of area under the curve, since this might have therapeutic significance. Line 361: There needs to be a bit more discussion of what clinical data might constitute state feedback. There are several small points worthy of addressing: In some places, the English is a bit awkward or stiff. It would be good were an experienced English-speaking editor help polish the prose. Lines 33-34: Should metabolism and excretion be added to the possible drug fates? Line 36: “incorporate” rather than “involve”. Line 44: It might be worthwhile to define “per-os” Line 54: “… in those, which … ” should read “… in those that …” Line 79: “ … cannula was taken out the sample … blood … ” should read “cannula a 2 ml sample of blood was withdrawn…” Line 8: “solubility” not “solvability” Figure 1 caption: Define GIT. It is defined on line 106, but the caption may be read before that line. Line 93: Define “A” and “c”. Line 122: “In contrast” instead of “Contrary”. Following Equation 6. Give “F” and “g” names Equation 9 and following. Are the four significant digits justified? I think so, but given the noise levels, it may not be meaningful. Line 193: “Remark, that…” should read “Note that…” Equation 16. Six digits? Line 245: “After elapsing the dosing interval ?=6 h …” is confusing. Is what was meant “At the end of the first dosing interval (T=6h),…” Line 252: I’m not sure how k_i = 0.3 translates to k1 = six hours, or such. Lines 287-294: The three sentences that read “The roots of its characteristic polynomial are .. circle, what implies that the closed-loop control is stable” are redundant and could be combined in either a table or other grammatical construction to not bore the reader with repeated statements that differ only by the values of the roots. Line 433: “completely safe” but also quite slow. Line 439. What might constitute “suitable state feedback”?

Author Response

See the attached document

Reviewer 4 Report

Dear authors,

the manuscript does not fit well the scope of the journal; whilst it is fairly structured  it contains many flaws in terms of design and methodology. The abstract is too narrative and does not present data sufficiently; the proof of principle is not obvious; the title is unclear; the novelty and pertinence of your study should be better highlighted in introduction; you must provide ethical review board approval number for study in animals; pharmacokinetics data should be presented and statistics applied; theorical and observed data should be validated; overall I found the manuscript and work insufficient for publication. The area should be pharmaceutical sciences and not really Bioengineering.

best,

the reviewer

Revisions needed

Author Response

See the attached document

Round 2

Reviewer 1 Report

The article has been improved.

Minor changes required in English language.

Author Response

The English editing has been performed.

See the tracked changes (Word software) in the manuscript.

Reviewer 4 Report

Dear authors,

Thanks for your answers and corrections. The title still must mention: "...b ased on  Mathematical Models or Computational Models"

Otherwise the manuscrit has been now much improved 

Best,

The Reviewer

Fine. Minor corrections needed

Author Response

The title has been changed

The English editing has been performed.

See the tracked changes (Word software) in the manuscript.